# Nanostructured Antibiotics and Their Emerging Medicinal Applications: An Overview of Nanoantibiotics

**DOI:** 10.3390/antibiotics11060708

**Published:** 2022-05-25

**Authors:** Shreya Modi, Gajendra Kumar Inwati, Amel Gacem, Shahabe Saquib Abullais, Rajendra Prajapati, Virendra Kumar Yadav, Rabbani Syed, Mohammed S. Alqahtani, Krishna Kumar Yadav, Saiful Islam, Yongtae Ahn, Byong-Hun Jeon

**Affiliations:** 1Department of Microbiology, Shri Sarvajanik Science College, Mehsana 384001, India; shreyamodi20@yahoo.in (S.M.); principal@sssc.edu.in (R.P.); 2Department of Chemistry, HVHP Institute of Post Graduate Studies and Research, Sarva Vishwavidyalaya, Kadi 382715, India; 3Department of Physics, Faculty of Sciences, University 20 Août 1955, Skikda 21000, Algeria; gacem_amel@yahoo.fr; 4Department of Periodontics and Community Dental Sciences, College of Dentistry, King Khalid University, Abha 62529, Saudi Arabia; sshahabe@kku.edu.sa; 5Department of Microbiology-Biosciences, School of Liberal Arts & Sciences, Mody University, Laxmangarh, Sikar, Rajasthan 332311, India; yadava94@gmail.com; 6Department of Pharmaceutics, College of Pharmacy, King Saud University, P.O. Box 2457, Riyadh 11451, Saudi Arabia; rsyed@ksu.edu.sa (R.S.); msaalqahtani@ksu.edu.sa (M.S.A.); 7Faculty of Science and Technology, Madhyanchal Professional University, Bhopal 462044, India; envirokrishna@gmail.com; 8Civil Engineering Department, College of Engineering, King Khalid University, Abha 61421, Saudi Arabia; sfakrul@kku.edu.sa; 9Department of Earth Resources & Environmental Engineering, Hanyang University, 222-Wangsimni-ro, Seongdong-gu, Seoul 04763, Korea; ytahn83@hanyang.ac.kr

**Keywords:** nanomaterials, nanocomposites, antibiotics, antimicrobial agents

## Abstract

Bacterial strains resistant to antimicrobial treatments, such as antibiotics, have emerged as serious clinical problems, necessitating the development of novel bactericidal materials. Nanostructures with particle sizes ranging from 1 to 100 nanometers have appeared recently as novel antibacterial agents, which are also known as “nanoantibiotics”. Nanomaterials have been shown to exert greater antibacterial effects on Gram-positive and Gram-negative bacteria across several studies. Antibacterial nanofilms for medical implants and restorative matters to prevent bacterial harm and antibacterial vaccinations to control bacterial infections are examples of nanoparticle applications in the biomedical sectors. The development of unique nanostructures, such as nanocrystals and nanostructured materials, is an exciting step in alternative efforts to manage microorganisms because these materials provide disrupted antibacterial effects, including better biocompatibility, as opposed to minor molecular antimicrobial systems, which have short-term functions and are poisonous. Although the mechanism of action of nanoparticles (NPs) is unknown, scientific suggestions include the oxidative-reductive phenomenon, reactive ionic metals, and reactive oxygen species (ROS). Many synchronized gene transformations in the same bacterial cell are essential for antibacterial resistance to emerge; thus, bacterial cells find it difficult to build resistance to nanoparticles. Therefore, nanomaterials are considered as advanced solution tools for the fields of medical science and allied health science. The current review emphasizes the importance of nanoparticles and various nanosized materials as antimicrobial agents based on their size, nature, etc.

## 1. Introduction

Heavy metals, such as copper, gold, silver, and others, have been employed as antibacterial agents over the last few decades owing to their nature and properties [1]. Considering their changes in oxidation state, many metal ions have the ability to denature and inactivate microorganisms, enzymes, and proteins. Metallic nanoparticles, which are much smaller in size, can easily enter the bacterium and kill it by disrupting its organelles. Because of their small size, electrical structures, and surface properties, nanomaterials have sparked a significant amount of attention in the last quarter. As a result, modified nanostructures are being extensively researched in the fields of nanomedicine, environmental remediation, and optoelectronics. The inorganic nanoparticles in metals (Ag, Cu, Au, etc.) and metal oxide (ZnO, CuO, TiO_2_, etc.) varieties are employed as antimicrobial agents [2,3,4]. Furthermore, several organic nanostructures and microstructures have antimicrobial capabilities due to their polymeric nature, wide surfaces, biocompatibility, and low toxicity, such as chitosan, dendrimers, liposomes, lignin, and cyclodextrin [5].

Infectious diseases induced by pathogenic bacteria are the primary cause of mortality globally and therefore constitute a persistent public health hazard across all nations [6]. Bacteria were identified as the first living organisms on the planet, and they have mutated to become resilient. The development of antibiotics is acknowledged as one of civilization’s greatest clinical breakthroughs during the modern period. From 1930 to 1962, roughly 20 novel antibiotics were developed to combat various bacterial infections. Furthermore, given the dynamic nature of emerging communicable disease, the pharmaceutical firms’ hunt for breakthrough molecules with efficient antimicrobial properties is becoming exceedingly challenging [7]. Approximately 700,000 people die each year as a result of incorrect antibiotic use, which builds resistance to normal therapies [8]. Several investigations have demonstrated conclusive evidence that antibiotic misuse has culminated in the emergence of multi-drug-resistant pathogenic organisms. Super bacteria, pathogens that have been shown to be resistant to almost all antibiotics, have recently evolved as a result of drug misuse [9].

It has been established that using nanomaterials as antibacterial agents does not result in the development of resistance to conventional antibiotics after a specific period of time. However, because nanoparticles are smaller in size, a specific amount (dose) of nanoparticles is required to create an effective antibiotic agent, and, thus, functionalized nanoscopic antibiotics have already been developed in relation to traditional antibiotics predicated on elemental compositions or stoichiometric ratios [10,11]. Nanoantibiotics have been reported to be more efficient, durable, and less toxic, depending on the manufacturing procedure and experimental settings. The current study focuses on recent advances in the realm of the antibacterial activity of metallic and nonmetallic nanoparticles and microparticles. This study also aims to investigate the antimicrobial activity of several classes of nanomaterials and structurally modified nanostructures. Organic (dendrimers [12], liposomes [12], functionalized chitosan [13]) and inorganic nanomaterials, such as surface-modified silver, gold, zinc oxide, titanium dioxide nanostructures, and copper nanocrystals, are said to have antimicrobial functionalities in biomedicine [9,11,14,15,16].

## 2. Physical Methods to Determine Antimicrobial Effects

Various physical and chemical approaches are widely used to control the growth of microorganisms. Some of the approaches are represented below, in Table 1.

## 3. Development of Antibiotic Resistance

### Limitations of Conventional Antibiotics

Innovation in modern antibiotics has slowed noticeably in the twenty-first century. However, four novel antibiotics were produced and licensed in the US market between 2008 and 2012, whilst 16 antibiotics were registered between 1983 and 1987 [22]. During the last forty years, no new antibiotics against Gram-negative bacteria have been produced; therefore, resistance to antibiotics has become a global concern, with record death rates and cases of fatal poisoning [22].

Nanotechnology is now widely regarded as a promising tool for changing the physical and chemical properties of a broad range of hybrid compounds in order to produce effective antibiotics [23]. Due to their large surfaces and narrow dimensions, several ranges of nanomaterials are noteworthy for their effective antibacterial properties. At quite low doses, nanoparticles with diverse morphologies and surface changes remain incredibly effective. As a result, nanomaterials can be utilized as powerful solutions, or antibiotics, to prevent infections caused by bacteria. Table 2 includes a comparison of traditional antibiotics and nanoantibiotics to help readers grasp the differences.

Evidently, numerous antibiotics have been discovered, and they have improved the standard of living by exhibiting effectiveness against microbial infections and establishing prospective medical interventions. Sir Alexander Fleming developed an antibiotic in 1928, and Florey and Chain sought to isolate reactive molecular component, known as penicillin, which were utilized as anti-bacterial agents; Penicillin was then commercially produced [25].

The history of antibiotics reveals that susceptibility in organisms evolves promptly following the introduction of antibiotics with a specific ingredient through industry. Consequently, pharmaceutical companies are not provided with sufficient incentives to develop new antibiotics [26]. Bacteria with different resistance genes are antibiotic-resistant. Such bacteria are referred to as “superbugs” because they can cause infections that are immune to conventional antibiotics. One survey revealed that more than 2.8 million antibiotic-resistant infections occur in the United States per year, killing 35,000 people (CDCP 2019). According to the European Centre for Disease Prevention and Control, bacteria-resistant disorders harm approximately 33,000 individuals in Europe annually [26]. Bacteria develop resistance in various ways (Figure 1).

## 4. Nanoparticles as Antimicrobial Agents

According to the World Health Organization, despite their limited sizes and specificity for microbes, some metal-based nanomaterials have proven to be beneficial against a wide range of infections [27]. Nanoparticles can serve as antibacterial agents on their own or as supplements for regular antibiotics; in either instance, they are termed “nanoantibiotics” (nAbts) [28]. NanoantibiotIcs is another term for nanosized antibiotic molecules that are encased with engineered nanoparticles or antibiotics manufactured artificially by keeping one dimension in the region of 100 nm [28]. Mamum et al. (2021) recently demonstrated the importance of nanoantibiotics as superior medicines to reduce the amount of drug-resistant bacteria in the treatment of diseases [29]. According to Mamun et al. “nAbts exemplify a potential Trojan horse technique to overcome antibiotic resistance processes”. Metal-based nanomaterials are frequently found to exert non-specific negative bacterial effects, such as the fact that they do not bind to a specific receptor in the bacterial cell, making it more difficult for bacteria to develop resistance (Figure 2) and extending the antibacterial field [28]. A wide range of nanostructures, including Au, Ag, CuO, TiO_2_, MgO, and ZnO, are projected to become antibacterial nanosystem alternatives [30,31,32].

### 4.1. Synergistic Effect (Antibiotics within Nanoparticles)

The influence of synergism is associated with the formation of exceptionally active hydroxyl radicals, as well as alterations in defensive cellular functions and anti-biofilm efficacy. It is associated with the use of antibiotics in connection with nanostructures, which is notably effective in increasing antibiotic functionality; this contrasts with the impact of the use of antibiotics in clinical practice, which minimizes antibiotic doses, exposure time, and bacterial resistance [33]. The majority of harmful microorganisms are resistant to conventional antibiotics. Nanoparticles are affixed to antibiotics to augment their potency. NPs fight pathogenic germs through many pathways, which are activated concurrently by nanoparticles and antibiotics. The significance of these contemporary approaches is that even if microorganisms have a large number of mutant genes, NPs may help to minimize bacterial resistance [32]. Research indicates that employing silver nanoparticles into certain drugs successfully increases the cumulative effects of antibiotics such as cefuroxime, azithromycin, fosfomycin, cefoxime, and chloramphenicol against *Escherichia coli*. Although the antibacterial efficacy of silver nanoparticles in conjunction with oxacillin and neomycin was reported to be lower against *Staphylococcus aureus* when contrasted with antibiotics alone, the conjunction of zinc oxide nanoparticles within antibiotics improved the antimicrobial efficacy [31]. On the other hand, following green-route synthesis employing *Allium sativum* extract to develop silver nanoparticles was also studied. The results demonstrated that the combined antibacterial activity of Ag NPs and cephalothin, cephem omycin, and cefazolin is an innovative and remarkable technique for drug carrier enhancement, which can be used in biomedical nanodevices. Cephem drugs are more expensive than other antibiotics; therefore, employing these nanoparticle combinations could also decrease the use of antibiotics, lowering their cost and their detrimental consequences [33]. Bankier et al. [34] performed research to evaluate the potential impact of several metallic nanoparticles on *Staphylococcus aureus* and *Pseudomonas aeruginosa*, either with or without antibiotics. The nanoparticles included tungsten carbide (WC), silver nanoparticles (Ag NPs), and copper nanoparticles (Cu NPs). Essentially, when the nanoparticles were mixed with antibiotics, the antimicrobial effects significantly improved compared to individual nanomaterials. This has also been reported by several other researchers in the field of nanomedicine and drug delivery.

### 4.2. Metal Nanoparticles (Inorganic Nanoparticles)

Metal and metal-oxide nanomaterials appear to be good inorganic nanostructures that have performed admirably as antibiotic resistance therapies. Nanomaterials act in different ways to antibiotics, indicating their efficacy against pathogens that have already acquired immunity. Furthermore, nanoparticles target a wide range of biomolecules, which affects the genesis of antibiotic strains [35,36].

#### 4.2.1. Antimicrobial Role of Silver Nanoparticles 

Ag NPs have antibacterial, antifungal, and antiviral properties. Ag NPs have the ability to pass through bacterial cell walls, modifying the cellular structure and, evidently, inflicting cell injury. Due to the larger exposed sites and narrow diameters of nanomaterials [36], when Ag NPs interact with bacteria, they accumulate at the membrane and form complexes, creating abnormalities that lead to cell death [37]. It is thought that Ag NPs release silver ions perpetually. Pei et al. [38] created Ag nanostructures with a nano range of 6–45 nm using *Coptis chinensis* (CC). The investigators assessed AgNPs against *Bacillus subtilis*, *Staphylococcus aureus*, *Pseudomonas aeruginosa, Klebsiella pneumonia,* and *Aspergillus niger*. The results revealed that the Ag NPs were exceptionally reactive with the *B. subtilis* and had a notably weaker impact on the *A. niger* at increasing concentrations (25, 50, 75, and 100 L/mL). Throughout the incubation phase, the AgNPs were found to carry a rapid generation of diffusible suppressive species from bacterial membranes.

Ag NPs of varied sizes (1–100 nm) interact with bioactive components to limit microorganism proliferation. By generating superoxide radicals, Ag NPs inhibit the growth of bacterial enzymes and induce cytotoxicity (ROS). Silver has also been found to serve as a buffer between thiol-containing molecules, causing permanent agglomeration. (Figure 3) [39,40] Many other substances, particularly DNA, peptides, and bio-molecules, have been identified as targets of the ions identified in the lethal bacterium. Antibiotics attempt to regulate particular elements of the bacterial life cycle, but silver ions can attach to any high-affinity component, implying that several bacterial cell mechanisms are disrupted, leading to cellular death. Further, silver ions are believed to bind non-specifically to a wide variety of sites and disrupt several aspects of cell metabolism simultaneously, eventually resulting in cell death [33].

#### 4.2.2. Gold Nanoparticles (Au NPs)

Gold nanoparticles have been used in a variety of applications, ranging from engineering to medical. Since gold nanoparticles are biocompatible, they have a greater potential for use in anticancer [41] and antibacterial drugs. MacDonald et al. [42] used a simple procedure followed by a thiol-functionalized AuNPs nanosystem to adjust the antibacterial surface structures to disrupt the bacterial cell in lighter and darker settings. The approach was examined and found to be significantly more effective against bacteria and other noxious microbiological entities. Gnanamoorthy et al. (2022) demonstrated the synthesis of 15-nanometer, average-sized AuNPs employing leaf extracts of Bauhinia tomentosa Linn. The synthesized AuNPs were assessed for their antimicrobial activity against *E. coli* and *Staphylococcus aureus* by the disk diffusion method. The disc diffusion method was used to test the antibacterial activity of the synthesized AuNPs against *E. coli* and *Staphylococcus aureus*. The results demonstrated that the plant-mediated AuNPs had greater antibacterial efficacy against both microorganisms [43].

### 4.3. Copper Nanoparticles (Cu NPs)

Copper is a ubiquitous metal that is also a fundamentally important mineral in the majority of life forms. Copper nanoparticles are employed in a multitude of scenarios, notably electrochemical sensors, optoelectronic devices, solar panels, and paints and varnishes [44]. Parikh et al. [31] employed Datura leaf extract to synthesize copper nanoparticles. The copper NPs outperformed standard chloramphenicol in antibacterial activity versus *E. coli*, *Bacillus megaterium*, and *Bacillus subtilis*, demonstrating that copper NPs can be used as antimicrobial agents instead of antibiotics. Chitosan was used as a stabilizer to inhibit the deposition and fast oxidation of copper nanoparticles with diameters ranging from 2–350 nm that were synthesized chemically. The antibacterial and antifungal activity of chitosan-copper NPs with different stoichiometric ratios (0.05 wt%, 0.1 wt%, 0.2 wt%, and 0.5 wt%, and chitosan) versus methicillin-resistant *Staphylococcus aureus, Bacillus subtilis*, *Pseudomonas aeruginosa,* and *Salmonella choleraesuis* has been studied. The 0.5 wt% combination demonstrated the greatest zone inhibition [45]. Kruk et al. [46] utilized hydrazine in an aqueous SDS solution to form Cu nanomaterials accompanied by Cu salt reduction. The Cu NPs were tested for antibacterial activity versus Gram-positive bacteria, including both conventional and clinical strains, namely methicillin-resistant *Staphylococcus aureus*, and antifungal activity towards *Candida* species. The results demonstrated that the as-prepared Cu NPs were more efficient as antimicrobial compounds than Ag nanoparticles, as well as some antibiotics [35].

### 4.4. Zinc Oxide Nanoparticles (ZnO NPs)

Transitional metals, such as zinc and iron, as well as their derivatives, are vital elements that play a pivotal role in the catalytic performances of a range of enzymes in the body and are widely dispersed across its tissues [24,47]. Gram-positive bacteria include *Staphylococcus aureus, Staphylococcus epidermis, Bacillus subtilis, Bacillus cereus, Listeria monocytogenes,* and *Escherichia faecium*. Gram-negative bacteria include *Pseudomonas aeruginosa, Escherichia coli, Klebsiella pneumoniae,* and *Salmonella* sp. [48]. Furthermore, *Acinetobacter baumannii* is a multi-drug-resistant opportunistic bacteria that primarily induces respiratory and urinary tract infections. Carbapenems, which belong to the beta-lactam group of antibiotics, have been found to be the most efficient medicines against *A. baumannii* so far; however, the progression of bacterial resistance to this antibiotic could result in extreme levels of fatality. To address this serious issue, scientists developed metal-based oxides, such as ZnO NPs, and observed them as prospective compounds to generate rapid reactive oxygen species, in order to increase membrane lipid peroxidation, resulting in the membrane leakage of reducing sugars, DNA, and proteins, as well as limiting cell survival. Researchers also proved that ZnO-NPs could be manufactured as novel anti-*A. baumannii* drugs [49]. Since semiconducting nanomaterials have the electrical properties necessary to function as antibacterial agents, they are under investigation as both pure and doped semiconductors. Inwati et al. [24] developed rare ion (Rb)-doped ZnO and utilized it as an antibacterial agent (Figure 4).

The antibacterial activity of synthesized nanostructures can be defined in terms of microbe cell destruction or plasma depletion after ion implantation. Furthermore, nanoparticle strikes on cellular components and cell dissociation in grown cells may affect bacterial routine cellular metabolism, leading to cytoplasmic discharge and bacterial cell death, with an unexpected rise in the inhibiting system. The more Rb were available, the more irregularities, such as oxygen vacancies, were incorporated into the ZnO frameworks. ZnO NPs with Rb deficiency generated pairs of electron-holes (e^−^–h^+^) during the VB-to-CB transition. A bandgap of 3.5 eV was detected in the ZnO NPs, yielding into holes and free electrons on the VB and CB, respectively. This hole-and-electron pair had a considerable impact on the antibacterial tests [24].

### 4.5. Titanium Dioxide Nanoparticles (TiO_2_ NPs)

Titanium dioxide (TiO_2_) NPs are among the most frequently researched nanomaterials for antimicrobial applications due to their unique properties, including bactericidal photocatalytic activity, security, and self-cleaning characteristics [50]. They offer significant potential as bactericidal and fungicidal agents in food packaging and containers [49]. When exposed to light, TiO_2_ NPs generate reactive oxygen species (ROS) with high oxidizing potential, leading to a shift in band-gap in the open atmosphere (O_2_) [51]. Arora et al. [52] investigated the role of TiO_2_ NPs in controlling the growth rate of *Pseudomonas aeruginosa* isolated by the endotracheal tract, pus, alveolar lavage, and sputum. They reported that exposing TiO_2_ nanoparticles to UV light for 60 min dramatically improved their antibacterial activity against *P. aeruginosa* with multiple drug resistance (MDR). Cefoxime’s potency was also demonstrated to be increased when coupled with UV-irradiated TiO_2_ NPs for 60 min. Combinational studies on the activity of TiO_2_ against pathogens emphasize its wider implications in clinical diagnostics as a strategy to combat the increasing problem of antibiotic resistance.

Semiconducting materials, such as TiO_2_, have holes that split H_2_O molecules into OH^−^ and H^+^ free ions, with freed electrons interacting with soluble oxygen ions. Molecular oxygen’s anionic superoxide (O_2_) radicals eventually caused the formation of hydrogen peroxide anions (HO_2_^−^) and H_2_O_2_. The H_2_O_2_ produced penetrated the cell membrane and caused damage to it. Consequently, the rate of H_2_O_2_ formation increased as the number of electron-hole pairs increased [53].

### 4.6. CuO Nanoparticles (CuO NPs)

CuO has recently been established as a robust p-type nanomaterial (1.2-electronvolt bandgap) for different applications due to its easy distribution and manufacture, better heat resistance, chemical inertness, and wider optical absorptivity. The morphological features of CuO (cupric oxide) or Cu_2_O (cuprous oxide) nanomaterials can be modified by adjusting their size or shape, such as cylindrical, pyramidal, 1-D or 2-D, nanowires, and so on [16,53]. Many professionals have modified and investigated the interfacial, electrical, physical, and chemical properties of pure CuO by applying alkali, transitional, and lanthanide ions. CuONPs are among the most frequently studied nanomaterials for antibacterial effects due to the fact that their appropriate ionic radii involve numerous ions such as Zn, Cd, Ag, Ce, Ni, and Co dopants [54,55]. When a metal ion is introduced into a CuO substrate, the structure and morphology of pure CuO crystalline solids changes. It has been observed that the ionic radii of the dopant play an essential part in the atomic interaction between both the dopant and the host environment. Ultimately, during the dispersion of the NPs into the deionized water, the electric charge chemically reacts with the active hole (h^+^) and free electrons (e^−^). BY contrast, superoxide ions are formed when free electrons contact with soluble oxygen atoms [32]. Furthermore, the hole at the top of the valence band reacts with aqueous ions to produce OH entities, which subsequently interact with superoxide ions to form hydrogen peroxide through H^+^ ions. As a response, the hydrogen superoxide disrupts the usual metabolism of the bacterial cell nucleus, and causes cell death.

Gnanamoorthy et al. (2021) synthesized copper aminophosphate (X-CuAP) nanoparticles with improved electrochemical, photocatalytic, and biological characteristics. The researchers observed that the (en)-CuAP nanocrystals had greater reusability and enhanced photocatalytic and antibacterial properties [56].

In addition, Gnanamoorthy et al. (2020) described the hydrothermal synthesis of (Cr)-CuSnO3 nanoparticles and their application for photocatalytic activity and antibacterial characteristics. The synthesized NPs were evaluated and their specific characteristics were examined using modern instruments [57].

Gnanamoorthy et al. (2021) synthesized morphologically diverse varieties of SnO_2_ rods and examined them for antifungal and antibacterial activity. The SnO_2_ rods were found to have antimicrobial effects on *Enterococcus fecalis* and antifungal activity against *Candida albicans* [2].

## 5. Organic Nanoparticles

Organic-based carbon nanostructures and nanocomposites are mostly composed of organic materials. Non-covalent interactions transform organic nanoparticles, such as liposomes, micelles, dendrimers, and polymeric NPs, into desired forms. At specific temperatures, organic antibacterial agents have been found to be slightly less stable than inorganic materials. Numerous polymers, polysaccharides, liposomes, cyclodextrins, and nanomaterials present in natural substances and macromolecules have been exploited as vectors for metallic and non-metallic nanomaterials in the modern age of nanoscience [58].

### 5.1. Liposome

Liposomes provide a secure biological delivery mechanism for hydrophobic and hydrophilic medicines. Liposomal antibacterial medicines are typically administered intravenously [59]. Liposomes are critical for the encapsulation of various medicines and antibiotics for targeted administration. Liposomal polytene macrolide antibiotics, amphotericin B (AmB), and nystatin have been developed as injectable dosage forms by researchers. AmB-encapsulating PEG liposomes (PEG-L-AmB) with optimum lipid composition were found to be less lethal and more effective in a murine model of pulmonary aspergillosis than standard AmB formulations [60]. Furthermore, scientists have discovered that the liposomal encapsulation of immunomodulators that activate macrophages can be used to attempt to reduce the toxicity of these drugs while directing them to cells in mononuclear phagocyte organizations to promote nonspecific resistance to diseases [61]. Eid and Azzazy [62] investigated synthesized Ag NPs with dextrose capping enclosed in liposomes. Liposome processing parameters, including an elevated homogenizer or reverse-phase evaporation with an extruder, are time-consuming and, therefore, do not account for liposome size and morphology. High-pressure homogenizers can generate 50-nanometer liposomes; however, this method is incompatible with drug encapsulation in the same process. Researchers developed a new method for producing nanoliposomes (50 nm) that enables the possibility of the encapsulation of AgNPs using a rotary evaporator and ultrasound. The ability of liposomes to release AgNPs for an extended period of time was investigated. Liposome Ag NPs (LAgNPs) were tested for antibacterial activity against Gram-positive and Gram-negative bacteria (*Escherichia coli, Salmonella enterica, Staphylococcus aureus, and Pseudomonas aeruginosa*) [63,64,65]. However, nanoscale organic liposomes have considerable antibacterial activity against microorganisms, diseases, and fungi. Their biological applications, good biocompatibility, and antibacterial actions, as well as their ability to function as absorption boosters, account for their widespread use in research. Figure 5 depicts a method for loading silver nanoparticles onto the surface of a liposome.

### 5.2. Cyclodextrin

Cyclodextrin belongs to the cyclic oligosaccharide family and is mostly composed of -1,4-linked glucopyranose subunits; it is useful as a molecular complexation agent [66,67]. There are three categories of cyclodextrin: α-CD, β-CD, and γ-CD. These are made up of six, seven, and eight α-1,4-linked glycosyl units, respectively. CD is produced by the enzymatic degradation during the starch breakdown process through enzymatic degradation. CDs are distinguished by their tendency to form molecular complexes with a wide range of solid, liquid, and gaseous combinations. Nikolic et al. (2018) constructed a complex of biochanin A (BCA) with (2-hydroxypropyl)-β-cyclodextrin (HP- β-CD) used in ethanol solution to improve its solubility in polar solvents, such as water. They predicted that the solubility of BCA would be twofold better in 42% (*v*/*v*) ethanol solution after complexation with HP- β-CD. As a result, a microdilution procedure was used to determine the antibacterial efficiency of an equipped inclusion complex against *Escherichia coli, Klebsiella pneumonia, Candida albicans*, and *Aspergillus niger*. Complexation has been shown to have no effect on BCA’s antibacterial activity [68].

### 5.3. Dendrimers

Metals have a range of dendritic shapes, such as diverging centers, ending portions, architectural reinforcements, and building components and linkages. Dendrimers are acknowledged as good NP carriers for antimicrobial drug delivery due to their unique properties, such as their increased surface area and relatively small size. Dendrimers with high-molecular-weight functionalized components can exceed the antibacterial properties of the associated molecule. Antimicrobial targets must be selected with extreme prudence at all times. Since larger dendrimers cannot penetrate the cell membrane barrier, they may have difficulty reaching the antimicrobial target [69,70,71]. Figure 6 shows the antibacterial activities of amphiphilic dendrimers.

Four poly (aryl ether)-based amphiphilic dendrimers with diverse terminal spacers containing amines, ester, and hydrazide groups were synthesized, described, and assessed for their antibacterial activity and mode of action. The impact of the dendrimers’ morphological characteristics for membrane interaction was also determined using molecular dynamics simulations. The cytotoxicity of the amphiphilic dendrimers was subsequently investigated on mouse fibroblast cells to validate their membrane selectivity for prokaryotes [65]. The findings presented here can be applied to the development of amphiphilic substances with varying hydrophobicity and cationic charge for better self-assembly and bactericidal activity. Additionally, detailed mechanistic research reveals that amphiphilic dendrimer hydrophobicity adjustment is crucial for bacterial cell membrane disintegration [63,66,68,69]. In recent years, the rise in invasive fungal infections and the emergence of antifungal resistance have highlighted the need for innovative antifungal treatments. Pure and functionalized peptides have been confirmed as viable candidates for the formulation of alternative antimicrobial therapeutics through enhanced screens and subsequent optimization using a reasonable method [70].

### 5.4. Chitosan Nanoparticles

Chitosan is a polysaccharide made by chitin, which can be found in crustaceans, insects, and the exoskeletons of fungi. It is employed in a variety of biomedical applications owing to its unique properties, such as bioactivity, gelation, adsorption ability, low toxicity, and nonimmunogenicity. It can be employed in textile mills as an antibacterial fiber ingredient and as a surface-engineering assistant for cellulose, cellulose/polyester, and wool fibers [65]. Chitosan is positively charged and soluble in acidic-to-neutral liquids due to the acidic strength (pKa = 6.5) of its amino acid group. It also possesses antibacterial characteristics that are attributed to its polycationic composition. It is formed by the hydrogenation of amino groups at the C-2 carbon of glucosamine groups. The positively charged surfaces of chitosan amino units can bind to the negatively charged bacterial surface, disrupting the cell membrane and its permeability strength. By interacting with bacterial DNA, it also suppresses protein biosynthesis. Chitosan’s antibacterial activity is influenced by its physical and chemical characteristics, molecular mass, and the balance of protonated and unprotonated amino groups in its chemical makeup. Moderate chitosan is thought to have better antibacterial activity than chitosan oligomers; its efficacy increases up to 90% with increased deacetylation. Chitosan does have the downside of being a weak cellulose fiber binder, resulting in slower leaching from the fiber’s surface after repeated washing. To facilitate chitosan’s attachment to cellulose fibers, polycarboxylic acids (1, 2, 3, 4, butantetracarboxylic, and citric acids) and imidazolidinone-derived products are used as cross-linking agents. Furthermore, the sensitivity of quaternized chitosan has been increased by adding functional acryl amido methyl groups to the primary alcohol groups (C-6), resulting in the development of covalent linkages with cellulose in alkaline mediums. Furthermore, metal ions, such as copper, zinc oxide, silver, and other metallic oxide NPs, can be wrapped in chitosan. Since chitosan requires an acidic environment, employing chitosan NPs offers several advantages, including the ability to handle the transport of bioactive compounds and the potential to prevent the use of toxic organic solvents while generating NPs. This nanocomposite is commonly used in medicine administration with organic and inorganic frameworks, particularly for specialized biomedical applications. In addition, mixing chitosan with inorganic NPs is a feasible strategy for creating antibacterial compounds with enhanced functional and antimicrobial properties [72,73,74,75,76,77]. Similarly, the target bacterium and the MW of chitosan have a substantial impact on its antibacterial effect. Lower MW chitosan provided superior antibacterial efficacy against Gram-negative bacteria (*E. coli, Klebsiella pneumoniae*, and *P. aeruginosa*), while Gram-positive bacteria seemed to have the reverse effect (*S. aureus* and *S. epidermidis*). The interaction of chitosan or liposomes with bacterial cells, by contrast, is the fundamental reason for the increased research on their use as antibacterial agents. The electrostatic interaction of positively charged organic molecules with the negatively charged bacterial cell surface is hypothesized to encourage increased penetration and apoptosis initiation. The electrostatic interaction between positively charged chitosan and negatively charged bacterial membranes increases bacterial surface transparency, leading to intracellular component release and, eventually, bacterial apoptosis [78]. It has also been claimed that chitosan decreases enzyme function by chelating to trace metals, thereby reducing bacterial growth.

### 5.5. Lignin Nanoparticles

Lignin and lignin nanoparticles have emerged as effective and dependable drug delivery agents in the medical field in recent years. They have also been employed for their direct antibacterial effects. One significant advantage of lignin nanoparticles is that they are abundant in nature, inexpensive, and environmentally beneficial. This is because lignin is mostly derived from plants, and when it is derived from plant waste, it is both cost-effective and ecologically sound. Numerous reports in the literature discuss the use of lignin as a drug-carrier molecule. Lintinen et al. (2019) discovered that lignin nanoparticles are highly effective against Gram-positive bacteria. The researchers also created a colloidal solution of silver and lignin and employed it as an antimicrobial agent [79]. The antimicrobial property of lignin, lignin-derived chitosan, and cellulose composites was reported by Alzagameem et al. (2019). The researchers used these nanocomposites to disrupt the biofilms generated by nanoparticles from both Gram-positive bacteria (*Bacillus thermosphacta*) and Gram-negative bacteria (*Pseudomonas fluorescens*). The developed nanocomposites exhibited about 30% efficiency for food-spoilage-causing bacteria, i.e., *P. fluorescens*, while its activity was only 5% for *B. thermosphacta* [80]. Recently, in one approach, Yun et al. (2021) obtained lignin with strong antibacterial activity from bamboo kraft [81]. Lignin has also been employed as a sustainable antimicrobial filler in various studies. Yadav et al. (2022) described the importance of lignin and lignin nanoparticles in a variety of disciplines, such as medicine, electronics, and environmental remediation [82].

Further, due to their large size and high commercial costs, peptides have minimal therapeutic potential. Artificial peptides are being developed as therapeutically viable drugs as a result of this constraint. The regioselective surface modification of DNA-encoded amino acids to produce enantiomerically pure adapted amino acids has recently received attention. These changed amino acids make excellent scaffolding for the creation of new designs for physically short synthetic peptides with significant drug delivery features. The tripeptides His(2-Ar)- and Trp-His(2-Ar), consisting in C-2 arylated histidines, have significant antifungal action against Cryptococcus neoformans with specific selectivity. In vitro, one of the more active peptides, 12f (His(2-biphenyl)-Trp-His(2-biphenyl)), showed excellent efficacy towards C. neoformans (IC50 = 0.35 g/mL, MIC = MFC = 0.63 g/mL) along a selective index of >28 and two times better efficiency than amphotericin B (IC50 = 0.35 g/mL). Peptide 12f was proteolytically stable but had little hemolytic action. The combination of 12f with fluconazole and amphotericin B at subinhibitory concentrations was found to be synergistically effective at damaging the plasma membrane of C. neoformans [83]. Due to their exceptional properties, naturally occurring antimicrobial peptides (AMPs) have attracted considerable attention and are being researched as new, safe, and successful antifungal medications. The synthesis of enantiomerically pure and modified amino acids via the regio-selective functionalization of DNA-encoded amino acids is currently a subject of increasing interest. These modified amino acids serve as good scaffolds for the development of new structural classes of robust peptides, which can be used as antifungal therapeutics in the medical field [70].

## 6. Carbon Nanotubes as Antimicrobial Agents

Carbon nanotubes (CNTs) are a class of antibacterial agents [84]. The literature is rich with accounts of CNTs being employed as drug carriers. CNTs are desirable agents in the biomedical industry due to their high mechanical strength, substantial photoluminescent property, and high surface-area-to-volume ratio. CNTs have been extensively used as effective and harmless drug delivery agents for both metallic and organic antimicrobial agents. Furthermore, there are multiple examples of CNTs being employed alone or surface-functionalized with various antimicrobial agents. Furthermore, in various cases, CNTs have been coupled with a polymeric substance, which may have antibacterial properties. Such CNT-based composites were shown to retain the toxic properties of CNTs to microbes whilst promoting cell–CNT interaction(Table 3).

CNTs have a wide variety of antimicrobial activities. In comparison with commercial antibiotics, functionalized multi-walled carbon nanotubes (F-MWNTs) were produced as antimicrobial nanomaterials. The generated F-MWNTs were tested for antimicrobial properties against Gram-positive *S.aureus* and Gram-negative *E.coli.* For *E.coli* and *S.aureus*, the optimal concentrations for maximal inhibition and antibacterial function were observed to be 80 and 60 g/mL, respectively. F-MWNTs were reported to be 85% more effective against E.coli and 57% more effective against *S.aureus* compared to conventional antibiotics [85]. Some researchers have documented CNT antibacterial activity. For example, Maas et al. (2016) demonstrated that when anchored to the surface, CNTs may block the adhesion of microorganisms, particularly bacteria, and prevent the development of bacterial biofilm [86]. CNTs’ antibacterial activity has also been reported by Malek et al. (2016) and Yick et al. (2015). Kang et al. (2007) revealed that single-walled carbon nanotubes were employed directly for antimicrobial purposes against *E. coli*, which injured the bacterial cell membrane. Furthermore, multiple studies have shown that multiwall CNTs, not just SWCNTs, are powerful antimicrobial agents. MWCNTs with small diameters promote partitioning and partial penetration into the cell wall. Longer CNTs offer better antimicrobial activities due to their exceptional agglomeration with the bacterial cell [84,87,88]. CNTs have antimicrobial activity for the purification of water and pathogen control, according to Liu et al. (2018). CNTs, in general, cause mechanical stress to microbial or bacterial cells, resulting in further cell disruption and, eventually, the release of intracellular substances. Furthermore, these CNTs cause oxidative stress in microorganisms [89]. The main disadvantage of using CNTs as antimicrobial agents is that they cannot compete with traditional techniques targeting biofilm owing to the lesser-known toxicity profile of human beings [90]. CNT anti-microbial properties are affected by a variety of parameters, including the CNT diameter and length, the residual catalyst, the electronic structure, the surface functional groups, the surface chemistry, the microbe type and shape, as well as the microbe growth state [91,92]. Fullerenes are made up of carbon units organized in spherical clusters. They exhibit antimicrobial properties against a number of pathogens, including *Salmonella, Streptococcus *spp., and *E. coli.* It is believed that when the nanostructures are ingested by the microorganisms, the decrease in energy metabolism improves their anti-bacterial activity. Fullerene compounds are suspected to inhibit bacterial growth by interfering with respiratory systems.

Notably, new pharmaceutical items must pass through all the phases of clinical trials, and they must be approved by the relevant authorizing agencies prior to the final phase. The progression from phase I to phase V, as well as its approval, applies to all clinical products, whether conventional or nanomaterial. Since nanomaterials are still under development, the majority of nanocosmeceuticals, nanopharmaceuticals, and other clinically important nanoformulations are still under clinical study. Numerous instances of nanoformulations with antimicrobial abilities under clinical study are listed below. Bruinenberg et al. (2010) [93] stated that ciprofloxacin-loaded liposomes might be employed to treat Pseudomonas aeruginosa respiratory functions. The current medication is in clinical phase III and further developments are required before it can be approved. Ciprofloxacin-loaded liposomes are also in phase IIa and III trials for respiratory infections and cystic fibrosis, respectively. Furthermore, Amikacin is being tested in stage II and III clinical trials [94]. Other antimicrobial peptides are yet to be commercialized and are currently undergoing clinical trials. Antimicrobial peptides, such as mutacin 1140 (MU1140), lipohexapeptides 1345 (HB1345), avidocin and purocin, arenicin (AP139), arenicin (AP138), arenicin (AP114), and no-varifyn are in the preclinical phase of development against various Gram-positive, Gram-negative, and antibiotic-resistant bacteria [95,96,97]. Despite the fact that there is a long list of clinical studies, the main issue is their high cost and time frame. After all the clinical trials and approvals are concluded, it is hoped that nanoformulations or nanotherapeutics could soon gain in market share.

## 7. Conclusions

The widespread use of conventional antibiotics has increased resistance in most bacterial strains, which are behaving as superbugs. This is becoming a major global public health hazard, mandating the invention of innovative antibacterial and other defensive agents to address the threat. Due to their small size and high aspect-to-volume ratio, many nanomaterials are quite well tailored as effective antibacterial agents. Furthermore, whenever nanoparticles are mixed with scaffold materials, they demonstrate adequate biocompatibility. Antibiotics, in particular, show rapid action and are difficult to manufacture and isolate. By contrast, nanomaterials/nanostructures provide long-term antibacterial activity with very few negative impacts. CNTs and other organic nanoparticles have demonstrated antimicrobial properties. Given the large range of conceivable procedures for these various nanostructured materials and the potential increase in their real-world applications, it is realistic to foresee a rapid expansion in this research area. Due to the obvious advantages of nanoparticles and their improved capabilities, significant attempts have been made to use these nanomaterials as antimicrobial interaction features in medical devices and appliances, such as textiles and fibers.

## Figures and Tables

**Figure 1 antibiotics-11-00708-f001:**
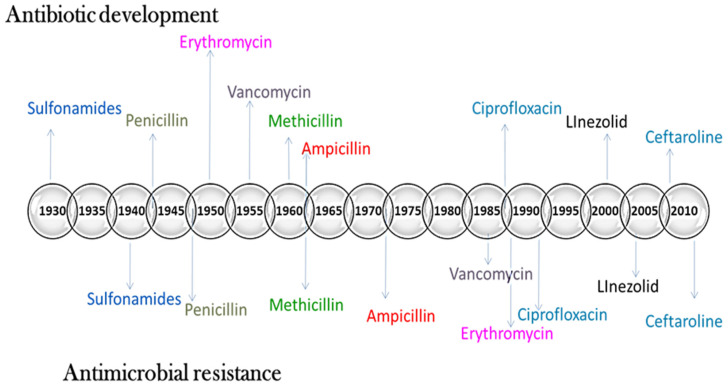
Mechanism of antibiotic resistance. History of antibiotic development and antibiotic resistance.

**Figure 2 antibiotics-11-00708-f002:**
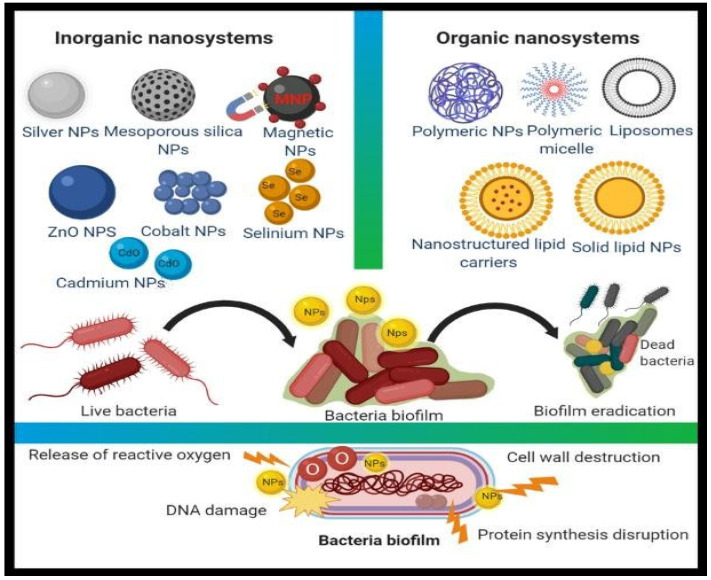
Different types of organic and inorganic nanomaterial for antimicrobial action [28].

**Figure 3 antibiotics-11-00708-f003:**
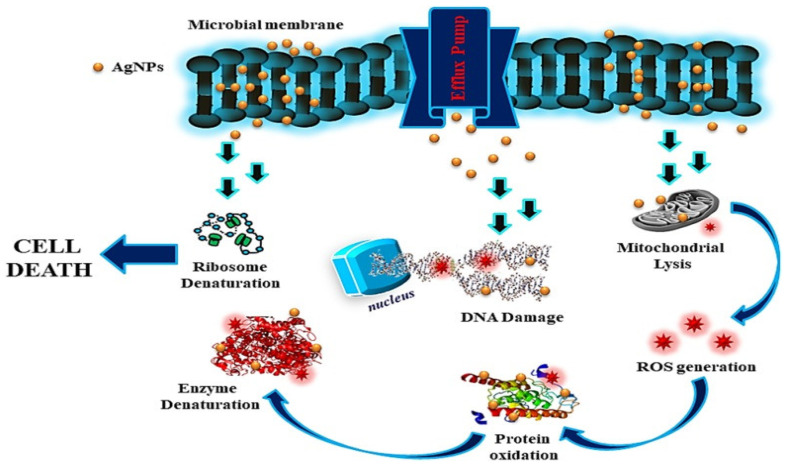
Mode of action of AgNPs Reprinted with permission from ref. [40]. Copyright 2018 Springer.

**Figure 4 antibiotics-11-00708-f004:**
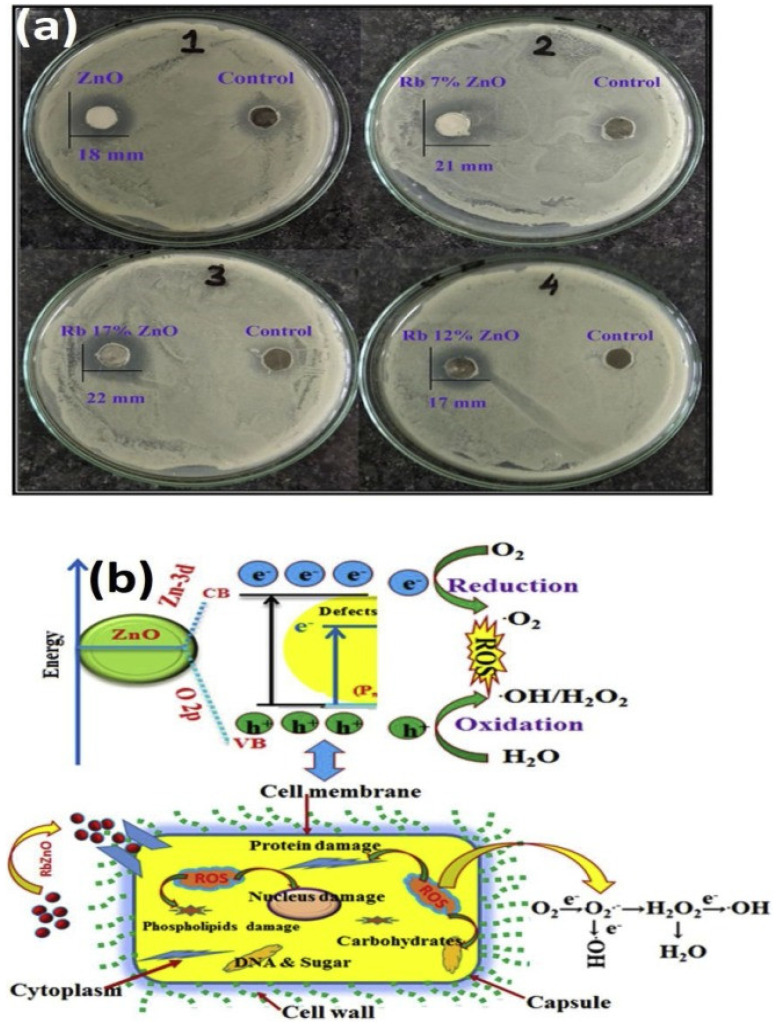
(**a**) Growth inhibition zone of *Bacillus subtilis* of pure ZnO and ZnO: Rb samples; (**b**) Schematic representation of cell death mechanism of *S. Bacillus* under the influence of ZnO: Rb. Antibacterial activities of Rb-doped ZnO NPs Reprinted with permission from ref. [24]. Copyright 2020 Elsevier.

**Figure 5 antibiotics-11-00708-f005:**
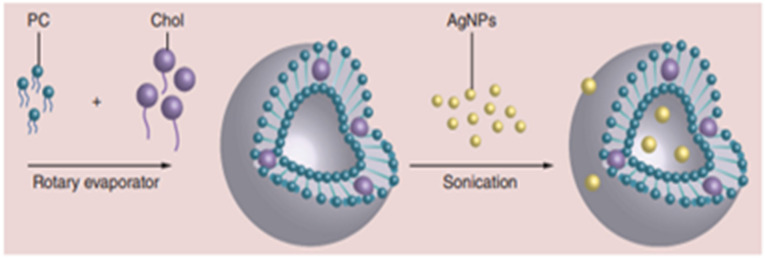
Preparation of liposome silver nanoparticles for antibacterial applications. This figure is taken with the permission of the ref [62]. Copyright 2014 Future Medicine Ltd.

**Figure 6 antibiotics-11-00708-f006:**
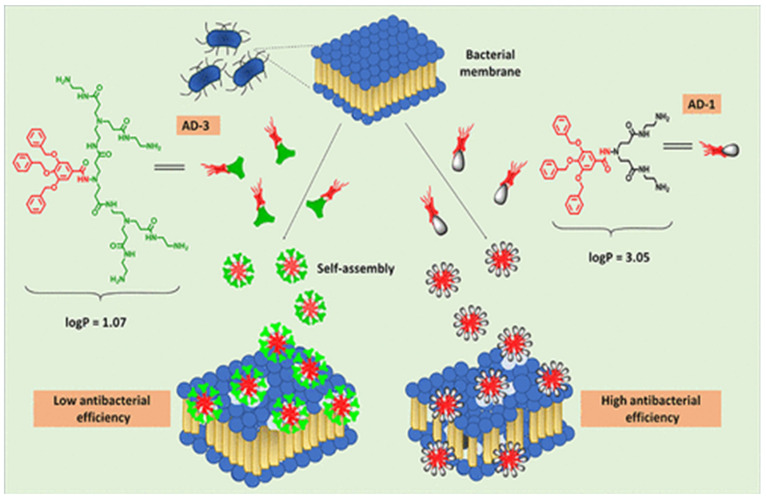
Functionalized poly (aryl ether)-based amphiphilic dendrimers for antibacterial activities [71].

**Table 1 antibiotics-11-00708-t001:** Antimicrobial physical agents.

Method	Characteristics	Mode of Action	Reference
Non-ionizing radiation (UV radiation)	260-nanometer UV range was studied as a prominent zone under 200–280-nanometer UV light	Induces thymine–thymine dimmers that subsequently inhibit the replication of DNA	[17]
Ionizing radiation	Electromagnetic radiation and particulate matter	Electron beams, as these are particulate in origin, generate high energy electrons, whereas gamma rays, which are electromagnetic, are used to sterilize a wide range of objects in seconds, including needles, bandage packs, edibles, and medications	[18]
Heat	Heat leads to oxidative effects and denaturation and coagulation of proteins.	Heat labile microbes are easily killed due to oxidative effects and protein denaturation	[19]
Dry heat	Generally used for sterilization purposes	Higher quantities of electrolytes cause irregular protein structures, radical formations, and lethal effects.	[19]
Humid hotness	More effective than dry heat Autoclaving is used at 121 °C for 15 min	The heat is under pressure, which increases its penetration power and kills the spores	[20]
Filtration	Different range of membrane filters is used, including earthenware filters, membrane filters, ultrafiltration, sintered glass, and nano-ranged filters or air filters	Separates microorganisms instead of killing them	[21]

**Table 2 antibiotics-11-00708-t002:** Difference between conventional antibiotics and nanoantibiotics Reprinted with permission from ref. [23]. Copyright 2011 Elsevier.

Conventional Antibiotics	Nanoantibiotics	References
Lost selective membrane permeability	Interrupt transmembrane transport	[22,23]
Antibiotics contain specific functional groups to inhibit biomolecules and their synthesis	Metal nanoparticles, such as ZnO NPs, Ag NPs ROS system damage cellular components, such as cell membrane/wall by adsorbing on the surface Inhibit enzyme and DNA synthesis Produce reactive oxygen species (ROS) that damage the cellular components Disturb energy transduction by interrupting transmembrane electron transport chain reaction, Release heavy metal ions with deleterious effects	[24]
Resistance to antibiotics is possible, as bacteria develop resistance genes	Offer resistance against genetic molecules in bacterial cells	[22,24]
Require high production costs and times	Require less time and feature lower production costs	[23]

**Table 3 antibiotics-11-00708-t003:** Organic, inorganic, and carbon-containing nanomaterials for targeted medicinal uses [84].

Nanomaterials	Antibiotics/Drugs	Target Bacteria/Diseases	References
Ag NPs	Ciprofloxacin, vancomycin Clotrimazole	VRE, MRSA MRSA, S. aureus,	[38,39]
Au NPs	Vancomycin, ampicillin	MRSA, MRSA, *P. aeruginosa*, *Enterobacter aerogenes, E. coli*	[42]
ZnO NPs	Ciprofloxacin, ceftazidime	MDRA. baumannii	[49]
Fe_3_O_4_ NPs	Ampicillin Ampicillin	*S. aureus* *E. coli, P. aeruginosa, MRSA*	
SWCNTs	Ciprofloxacin	*S. aureus, P. aeruginosa, E. coli*	[84,85]
Chitosan	Streptomycin Ciprofloxacin	*Listeria monocytogenes* *Uropathogenic E. coli*	[62]
Liposome	Pioglitazone (PIO), dexamethasone plus minocycline	Atherosclerotic plaques Orthopedic/dental implants	[59]
Exosome	Curcumin	Septic shock	[68]

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
