# Peer review of "Nanostructured Antibiotics and Their Emerging Medicinal Applications: An Overview of Nanoantibiotics"

_antibiotics, 2022, doi:10.3390/antibiotics11060708_

Round 1

Reviewer 1 Report

Reviewer comments (Antibiotics)

This manuscript describes “Development of Antibiotic Nanosystems: An Overview to-2 wards Structural and Surface Metamorphosis of Nanostructures”. This is an interesting review article on Antibiotic Nanosystems, however there are many places that need improvement before its publication. However, this is some major and minor issue in the current manuscript, and therefore this work need improvement. It can be considered for publication after addressing following concerns.

Major and Minor concerns:

  • Line 10-11 lines in abstract, “Bacterial strains impervious to antimicrobial agents such as antibiotics, now being used 10 have become genuine general medical issues that build the need to grow new bactericidal materials.” This is not proper statement and need to be rewritten.
  • Overall introduction need more improvement and need more information about background.
  • It would be better if authors can add one table and discussion on comparison between convectional antibiotics vs nano antibiotics for clarity.
  • Clarity in writing and its significance of summarized work is missing in this article. It would need major improvement in presentation of data. It would be better if authors can add tables for each section of organic, inorganic and Carbon nanotubes sections. Tables may contain particle involved, significance of work, year of publication and special comments related to work.
  • A section about clinical trials products or marketed products can be added to provide value of such work to readers.
  • Section 6. Carbon nanotubes as an antimicrobial agent, need more details and data. Authors need to cover more articles in this section.
  • All reference should in uniform pattern.  

Author Response

This manuscript describes “Development of Antibiotic Nanosystems: An Overview to-2 wards Structural and Surface Metamorphosis of Nanostructures”. This is an interesting review article on Antibiotic Nanosystems, however there are many places that need improvement before its publication. However, this is some major and minor issue in the current manuscript, and therefore this work need improvement. It can be considered for publication after addressing following concerns.

Response: Authors are highly thankful to the reviewer for observing the fundamental and technical errors. The given suggestions are made and the revised version is made by incorporating the reviewer's suggestions.

Major and Minor concerns:

  • Line 10-11 lines in abstract, “Bacterial strains impervious to antimicrobial agents such as antibiotics, now being used 10 have become genuine general medical issues that build the need to grow new bactericidal materials.” This is not proper statement and need to be rewritten.

Response: The statement has been corrected in revised version.

  • Overall introduction need more improvement and need more information about background.

Response: The corrections have been made in revised manuscript.

  • It would be better if authors can add one table and discussion on comparison between convectional antibiotics vs nano antibiotics for clarity.

Response: We have made the comparison between conventional antibiotics and nano antibiotics and added (Table 2) in the revised version.

Conventional antibiotics

Nano-antibiotics

Lost selective membrane permeability

Interrupt transmembrane transport

Antibiotics contain specific functional groups to inhibit biomolecules and their synthesis

Metal nanoparticles like ZnO NPs, Ag NPs ROS system damage cellular components like cell membrane/wall by adsorbing on the surface,

inhibit enzyme and DNA synthesis

produce reactive oxygen species (ROS) and reactive NO that damage the cellular components

disturb energy transduction by interrupting transmembrane electron transport chain reaction

releases heavy metal ions with deleterious effects (Huhand 2011).

Resistance to antibiotics is possible as bacteria develop resistance genes

Resistance is not possible

Require high cost and time for production

Require less time and lower cost for production

  • Clarity in writing and its significance of summarized work is missing in this article. It would need major improvement in presentation of data. It would be better if authors can add tables for each section of organic, inorganic and Carbon nanotubes sections. Tables may contain particle involved, significance of work, year of publication and special comments related to work.

Response: Corrections are made as per given comments

Table 3

Nanomaterials

Antibiotics/Drugs

Target Bacteria/Diseases

Ag NPs

Ciprofloxacin Vancomycin

Clotrimazole

VRE,

MRSA

MRSA, S. aureus,

Au NPs

Vancomycin,

Ampicillin

MRSA,

MRSA, P. aeruginosa, Enterobacter aerogenes, E. coli

ZnO NPs

Ciprofloxacin, ceftazidime

MDRA. baumannii

Fe3O4 NPs

Ampicillin

Ampicillin

S. aureus

E. coli, P. aeruginosa, MRSA

SWCNTs

Ciprofloxacin

S. aureus, P. aeruginosa, E. coli

GO

Lincomycin hydrochloride

Chloramphenicol

Gentamycin sulfate

E. coli, S. aureus

Chitosan

Streptomycin

Ciprofloxacin

Listeria monocytogenes

Uropathogenic E. col

Liposome

Pioglitazone (PIO),

dexamethasone plus minocycline

Atherosclerotic plaques

Orthopedic/ dental implants

Exosome

Curcumin

Septic shock

  • A section about clinical trials products or marketed products can be added to provide value of such work to readers.

Response: The investigation would look at the pharmaceutical supply chain structure holistically before reducing its focus to the antibiotics supply sector. After the drug has been validated, the production process begins with the preparation of building ingredients, accompanied by the production phase, which consists of two phases: manufacturing the Active Pharmaceutical Ingredients (API) and then the synthesis phase, followed by the labeling step and allocation to the target customer. To get a better understanding of the worldwide organization of the antibiotics to offer prospects, and because huge income equals big market share This study focuses on large pharmacological businesses.

Because of a paucity of statistical studies on individual company sales, revenues, and market shares of antibiotics, this study includes five other statistics (aspects) to assist characterize the shape of the antibiotics supply sector. The five factors evaluated in gathering data on the antibiotics market are as follows; The organization of the antibiotics industry's distribution network, worldwide antibiotic commerce (outsourcing and buying countries), antibiotic use, and pharmaceutical revenues (marked and generic).  Reports on the main players in the antibiotics supply network can be found on websites.

Antibiotics can be created in a variety of ways, according to the WHO Analysts surveyed (2018), including completely integrated enterprises that control the entire manufacturing line from raw materials to end customers, as well as being the market permission holder (MAH). Different businesses can be a part of the network as a MAH or a contract manufacturing and development organization (CMDO).

  • Section 6. Carbon nanotubes as an antimicrobial agent, need more details and data. Authors need to cover more articles in this section.

We have added recent research study for Carbon nanotubes as an antimicrobial agent

  • All reference should in uniform pattern.  

Response: Corrections are made.

Reviewer 2 Report

The manuscript presented for review concerns "nano-antibiotics" as poweful weapon against resistant microorganisms. In the first part of this manuscript, the authors described metallic/metal oxide nanoparticles as "nano-antibiotics", but the antimicrobial properties of these nanostructures have been showed many times. No consistent transmission of the antimicrobial activity of these structures was found in the manuscript. The selected literature items were cited but the authors did not show the advantage of this references. The antimicrobial properties of chitosan, liposomes etc. are shown in the next part of the manuscript, but it is also not explained why only this examples are presented. It seems that this manuscript does not have "a new look" at the problem of "nano-antibiotics" and should not be published in this form.

Author Response

The manuscript presented for review concerns "nano-antibiotics" as poweful weapon against resistant microorganisms. In the first part of this manuscript, the authors described metallic/metal oxide nanoparticles as "nano-antibiotics", but the antimicrobial properties of these nanostructures have been showed many times. No consistent transmission of the antimicrobial activity of these structures was found in the manuscript. The selected literature items were cited but the authors did not show the advantage of this references. The antimicrobial properties of chitosan, liposomes etc. are shown in the next part of the manuscript, but it is also not explained why only this examples are presented. It seems that this manuscript does not have "a new look" at the problem of "nano-antibiotics" and should not be published in this form.

Response: Authors are highly thankful to the reviewer for observing the fundamental and technical errors. The given suggestions are made and the revised version is made by incorporating the reviewer's suggestions.

The antimicrobial activity of nanoscale organic chitosan and liposomes over microbes, pathogens, and fungus is extensive. Its biomedical applications, good biocompatibility, and antimicrobial activities are the reasons for its considerable use in investigation, as well as the capacity to function as an absorption booster. The antibacterial action of chitosan is significantly influenced by the target bacterium as well as the MW of chitosan. For Gram-negative bacteria (E. coli, Klebsiella pneumoniae, and P. aeruginosa), lower MW chitosan had better antibacterial action, while Gram-positive bacteria had the opposite effect (S. aureus and S. epidermidis). However, the interaction of chiton or liposomes with bacterial cells is the primary reason for increased research on them as antibacterial agents. The electrostatic interaction of positively charged organic molecules with the negatively charged bacterial cell surface is thought to cause enhanced permeation and induction of apoptosis.

According to one study, the electrostatic interaction between the positively charged chitosan and the negatively charged bacterial membrane enhances the transparency of the bacterial surface, leading to intracellular component release and finally bacterial apoptosis [120]. It's also been proposed that chitosan reduces enzyme activity by chelating to trace metals, preventing bacterial development.

Reviewer 3 Report

The present manuscript includes a literature review regarding the antibacterial activity of nanomaterials. The work is not appropriate for publication since it contains numerous mistakes. Please consider the following observations.

  1. English grammar needs improvement. The meaning of some sentences changes due to grammatical mistakes. For instance, look at the following paragraph: From 1930 to 1962, increasingly than 20 new antibiotic classes have were developed, however, due to the dynamic nature of new pathogenic pathogens, the biopharmaceutical industry's identification of novel compounds with antibacterial activities has grown more difficult and demanding. The authors MUST at least use the software Grammarly to correct basic grammar mistakes.
  2. The work is not considered relevant due to its lack of novelty. Most of the information is very basic and already known.
  3. The title is not accurate. "Development of Antibiotic Nanosystems: An Overview towards Structural and Surface Metamorphosis of Nanostructures". An overview of the structural and surface metamorphosis of nanostructures? The contents of the work are not following the title. 
  4. Lines 38 to 41: " Among inorganic nanoparticles, both metallic form and their oxides form are used as antimicrobial agents. Metallic nanoparticles like Ag, Au, Cu, etc. have antimicrobial activities while metal oxides like titanium dioxide, zinc oxide, copper oxide, etc. have antimicrobial effects". What is the difference between antimicrobial activities and antimicrobial effects?
  5. Lines 64-65: "In addition to this, there is less chance of contamination of the nanomaterial product in comparison to the conventional antibiotics where the chances of contamination are very high". The asseveration is not necessarily true. It will depend on the production process and product manipulation of the user. 
  6. Line 74: Physical ingredients towards antimicrobial impacts. MUST say, physical methods.
  7. Figure 1 is not a figure, it is a diagram. It is not a graphic representation of how bacteria develop antibiotic resistance. 
  8. Figure 4 is not appealing for a review article.
  9. Figure 5 describes the antibacterial activity of silver NPs. Lixiviation of Ag ions is an important antibacterial mechanism of this material. Lixiviation of ions is not being considered in Figure 5. The manuscript is hampered by grammar mistakes. Also, some paragraphs are not linked to the context, for instance: The semiconducting materials like TiO2 holes split the H2O molecules into OH- and H+ free ions, with the liberated electrons reacting with the soluble oxygen ions. The anionic superoxide (O2) radicals created by the molecular oxygen finally resulted in the creation of hydrogen peroxide anions (HO2 - ) and H2O2. The H2O2 generated entered the cell membrane and caused harm to the cell membrane. As a result, it's worth noting that the rate of H2O2 creation rose as the number of electron-hole pairs was generated.

Author Response

The present manuscript includes a literature review regarding the antibacterial activity of nanomaterials. The work is not appropriate for publication since it contains numerous mistakes. Please consider the following observations.

Response: Authors are highly thankful to the reviewer for observing the fundamental and technical errors. The given suggestions are made and the revised version is made by incorporating the reviewer's suggestions.

  1. English grammar needs improvement. The meaning of some sentences changes due to grammatical mistakes. For instance, look at the following paragraph: From 1930 to 1962, increasingly than 20 new antibiotic classes have were developed, however, due to the dynamic nature of new pathogenic pathogens, the biopharmaceutical industry's identification of novel compounds with antibacterial activities has grown more difficult and demanding. The authors MUST at least use the software Grammarly to correct basic grammar mistakes.

Response: Grammar mistakes are made in the revised version of the manuscript.

2.The work is not considered relevant due to its lack of novelty. Most of the information is very basic and already known.

Response: The whole manuscript is revised by adding suitable content to the text.  

3.The title is not accurate. "Development of Antibiotic Nanosystems: An Overview towards Structural and Surface Metamorphosis of Nanostructures". An overview of the structural and surface metamorphosis of nanostructures? The contents of the work are not following the title. 

Response: Title has been changed, new title is as “Nanostructured Antibiotics and their Emerging Medicinal Ap-plications: An Overview of Nanoantibiotics”

4.Lines 38 to 41: "Among inorganic nanoparticles, both metallic form and their oxides form are used as antimicrobial agents. Metallic nanoparticles like Ag, Au, Cu, etc. have antimicrobial activities while metal oxides like titanium dioxide, zinc oxide, copper oxide, etc. have antimicrobial effects". What is the difference between antimicrobial activities and antimicrobial effects?

Response: Corrections are made as per suggestions.

5.Lines 64-65: "In addition to this, there is less chance of contamination of the nanomaterial product in comparison to the conventional antibiotics where the chances of contamination are very high". The asseveration is not necessarily true. It will depend on the production process and product manipulation of the user.

Response: The content is modified as per the given suggestion.

6.Line 74: Physical ingredients towards antimicrobial impacts. MUST say, physical methods.

Response: Corrections are made.

7.Figure 1 is not a figure, it is a diagram. It is not a graphic representation of how bacteria develop antibiotic resistance. 

Response: Corrections done by removing it.

8.Figure 4 is not appealing for a review article.

Response: Corrections done by removing figure 4.

9.Figure 5 describes the antibacterial activity of silver NPs. Lixiviation of Ag ions is an important antibacterial mechanism of this material. Lixiviation of ions is not being considered in Figure 5. The manuscript is hampered by grammar mistakes. Also, some paragraphs are not linked to the context, for instance: The semiconducting materials like TiO2 holes split the H2O molecules into OH- and H+ free ions, with the liberated electrons reacting with the soluble oxygen ions. The anionic superoxide (O2) radicals created by the molecular oxygen finally resulted in the creation of hydrogen peroxide anions (HO2 - ) and H2O2. The H2O2 generated entered the cell membrane and caused harm to the cell membrane. As a result, it's worth noting that the rate of H2O2 creation rose as the number of electron-hole pairs was generated.

Response: Authors have given a suitable example for metal and metal-oxides, the possible ionic species production during cell damage. It is well reported that the radical species are highly responsible for the imbalance of the metabolism inside the cell. And, thus we have explained based on electron-hole pairs and their role in further radical productions. The free-electron involve to produce the radicals and thus the recombination time delay for electron-hole pairs. Thus, we have discussed this concept by giving all possible free radicals formations.

Round 2

Reviewer 1 Report

Reviewer comments (Antibiotics)_revision

This manuscript describes “Nanostructured Antibiotics and their Emerging Medicinal Applications: An Overview of Nanoantibiotics”. Authors have improved manuscript after revision. However, this are still some major and minor issues in the current manuscript, and therefore this work needs some more improvement.

Major and Minor concerns:

  • Line 97-98, “During the years 2008-2012 only four new antibiotics are developed and approved for the 97 US market.” Authors need to provide update upto April 2022, not 2012. Therefore, authors need to revisit antibiotic discovery and update it with relevant information.
  • Table 2, Can authors provide reference for “Resistance is not possible” in case of Nano-antibiotics ? Authors also need to provide reference for “Require less time and lower cost for production” ? How is this possible? . This table also need referencing like table 1.
  • For Table 3 Organic, inorganic and carbon containing nanomaterials for targeted medicinal uses [84], authors need to add reference for every nanomaterials and its microbial strains. They cannot copy a table from some review and cite it here. Specific research article needs to be referenced accordingly. This table also need referencing like table 1.
  • Authors response for previous comment is completely out of topic. “A section about clinical trials products or marketed products can be added to provide value of such work to readers.” It is hard to understand what they are trying to say in their response. What need to be answered is that, is there any nano-antibiotic is under development/clinical trails? If yes, they need to add discussion about this as this review basically about nano-antibiotics.

Author Response

Please find the attached doc file for all comments and suggestions. 

We have responded point by point to all comments in the attached file.

Reviewer 2 Report

Thank you for answers.

Author Response

Thank You for giving the valuble comments and Acceptance.

Regards

Reviewer 3 Report

The authors DID NOT ATTEND REVIEWER's COMMENTS. Please look at the first recommendation on a previous version of the work:

  1. English grammar needs improvement. The meaning of some sentences changes due to grammatical mistakes. For instance, look at the following paragraph: From 1930 to 1962, increasingly than 20 new antibiotic classes have been developed, however, due to the dynamic nature of new pathogenic pathogens, the biopharmaceutical industry's identification of novel compounds with antibacterial activities has grown more difficult and demanding. The authors MUST at least use the software Grammarly to correct basic grammar mistakes.

Response: Grammar mistakes are made in the revised version of the manuscript. 

The authors are right!!!! There are MANY GRAMMAR MISTAKES!!!!

From ancient times only, numerous metallic particles were used as antimicrobial agents like copper, gold, silver, etc., due to their heavy metal nature [1]. Copper ions may denature the enzymes and proteins of the microbes and may inactivate them. With the advent of time, nanotechnology and nanoparticles being much smaller in size could easily access the entry into the microbes and may inactivate or kill them. No doubt, due to
the above-mentioned features, nanoparticles have gained huge attention in the last decade in medicine, environmental cleanup, electronics, etc., but are most widely used in medicine. Among inorganic nanoparticles, both metallic form and their oxides form are used as antimicrobial agents. Metallic nanoparticles like Ag, Au, Cu, etc. have antimicrobial effects while metal oxides like titanium dioxide, zinc oxide, copper oxide, etc. have antimicrobial effects [2-4]. Besides this, there are various organic nanoparticles and
microparticles also exhibits the antimicrobial effect, for instance, chitosan, dendrimers, liposomes, lignin nanoparticles, cyclodextrin, etc. these organic micros and nanoparticles may exert their antimicrobial effect due to their large polymeric size [5].

Difficult to understand if the small or large size is responsible for the antimicrobial effect!!!!!!!!

"From 1930 to 1962, increasingly than 20 new antibiotic  classes were developed, however, due to the dynamic nature of new pathogenic pathogens, the biopharmaceutical industry's identification of novel compounds with antibacterial activities has grown more difficult and demanding.

SEVERAL GRAMMAR MISTAKES!!!!!!

The authors include very few changes in the content of the manuscript. Lack of novelty and academic inconsistencies limit the acceptance of the present work. 

Author Response

The authors are thankful for these observations and suggestions. We have revised the manuscript by correcting language, and sentences with grammar particularly for a mentioned paragraph in the comments. And, the revised version is also modified by adding suggested text (response) as per another reviewer. Now, we hope that the current form of the revised article would be more suitable for acceptance.

Regards

Round 3

Reviewer 1 Report

Reviewer comments (Antibiotics)_revision 2

This manuscript describes “Nanostructured Antibiotics and their Emerging Medicinal Applications: An Overview of Nanoantibiotics”. Authors have addressed reviewers concerns in revision. New title of manuscript is better than previous one. However, there are still some minor issues in the current manuscript. It can be accepted after these minor improvements.

Minor concerns:

  • In this section [Antimicrobial peptides like Mutacin 1140 (MU1140), lipohexapeptides 1345 548 (HB1345), Avidocin and purocin, Arenicin (AP139) Arenicin (AP138), Arenicin (AP114) 549 and novarifyn against various gram-positive, grams negative and antibiotics resistant 550 bacteria is under preclinical phase. (95, 96, 97).], authors discussed only about antimicrobial peptides. However, there are discovered some antifungal peptides also and It also need to be added in subsequent discussion. Following references can be useful for antifungal peptides class and can be cited in manuscript. For antifungal peptides (Journal of medicinal chemistry,2017, 60(15), 6607-6621; Future medicinal chemistry, 2016, 8(12), 1413-1433.).
  • Acknowledgments and funding text need to be checked as both have similarity.
  • Authors followed multiple referencing pattern so it need to be corrected. All reference should in uniform pattern.

Author Response

Response: Authors are thankful for these valuable comments. We have added the antifungal part in the revised version as per your suggestions and also added suggested references. The grammar mistakes and sentence sequences with language corrections are been made in this revised version. The funding text is now mentioned only one time.

We hope that the revised version is much more suitable for acceptance in its current form for this journal now.

Regards

Reviewer 3 Report

I do acknowledge the efforts of the authors to try to improve the present work. Unfortunately, the manuscript still contains many grammar and academic mistakes. Please look at the following paragraph:

"From the above literature, it has been found that the utilization of nanoparticles as an antimicrobial agent will not develop resistance easily as in the case of conventional antibiotics after a certain period of time. Moreover, being smaller in size, a small dose of nanoparticles will be very effective as an antimicrobial agent. Moreover, it is very easy to surface functionalize nanoparticles in comparison to conventional antibiotics. Moreover, nanomaterial-based antibiotic synthesis is easier and more economical in comparison to traditional antibiotics [10]. Moreover, there is less possibility of pollution of the nanomaterial compounds where the possibilities of the contamination may be high, depending upon the production process and product manipulations of the customers. The pre-
sent review work emphasizes the recent progress in the field of antimicrobial activity of metallic and nonmetallic nanoparticles and microparticles. Moreover, another objective of this review is to explore antimicrobial activities and how they can be addressed using several nanomaterials and nanostructures. Antimicrobial applications of organic (dendrimers [11], liposomes [11], functionalized chitosan [12]) and inorganic nanomaterials 
such as surface-modified silver, gold, zinc oxide, titanium dioxide nanostructures, and copper nanocrystals have been documented in diverse investigations [9, 13-16]."

  1. Moreover, being smaller in size, a small dose of nanoparticles will be very effective as an antimicrobial agent. This statement is not true. The dose depends on the composition. For example, silver nanoparticles inactivate bacterial growth at lower doses than copper nanoparticles,
  2. The word moreover appears many times in the same paragraph.
  3. Moreover, there is less possibility of pollution of the nanomaterial compounds where the possibilities of the contamination may be high, depending upon the production process and product manipulations of the customers. Previously, I asked the authors to properly modify this sentence. Still, the meaning is incorrect. 
  4. I do not believe this work should be accepted. It is full of grammar mistakes and academic concepts.

I put another example included in the reviewed form of the manuscript:

The development of new antibiotics has dialed back observably in the 21st century. During the years 2008-2012 only four new antibiotics are developed and approved for the US market. While during 1983 to 1987, 16 antibiotics had been approved. As a matter of fact, no novel antibiotics have been developed in the last 40 years against gram-negative bacteria. The explanation behind that could be logical, business, and administrative. Resistance to antibiotics has turned into a worldwide risk for unprecedented death rate and lethal contamination [22].

  1. During the years 2008-2012 only four new antibiotics are developed and approved for the US market. Reference to sustain this asseveration is missing.
  2. While during 1983 to 1987, 16 antibiotics had been approved. As a matter of fact, no novel antibiotics have been developed in the last 40 years against gram-negative bacteria. Reference to sustain this asseveration is missing.
  3. Resistance to antibiotics has turned into a worldwide risk for unprecedented death rate and lethal contamination [22]. There are MORE RELEVANT REFERENCES in the literature to sustain this asseveration.

Zaman, S. B. et al., A review on antibiotic resistance: alarm bells are ringing. Cureus, 2017, 9(6).

Author Response

The authors are highly thankful to the reviewer for these valuable observations and comments. We have revised the manuscript as per the given suggestions and the revised version is submitted. Please find the point-by-point response in the attached file. 

Regards
